# The Contribution of Germline Pathogenic Variants in Breast Cancer Genes to Contralateral Breast Cancer Risk in *BRCA1/BRCA2/PALB2*-Negative Women

**DOI:** 10.3390/cancers15020415

**Published:** 2023-01-08

**Authors:** Alexey Larionov, Eleanor Fewings, James Redman, Mae Goldgraben, Graeme Clark, John Boice, Patrick Concannon, Jonine Bernstein, David V. Conti, Marc Tischkowitz

**Affiliations:** 1Department of Medical Genetics, National Institute for Health Research Cambridge Biomedical Research Centre, University of Cambridge, Cambridge, Cambridge CB2 0QQ, UK; 2School of Water, Energy and Environment, Cranfield University, Cranfield, Bedford MK43 0AL, UK; 3Division of Epidemiology, Department of Medicine, Vanderbilt University Medical Center, Nashville, TN 37232, USA; 4Genetics Institute and Department of Pathology, Immunology and Laboratory Medicine, University of Florida, Gainesville, FL 32610, USA; 5Department of Epidemiology and Biostatistics, Memorial Sloan-Kettering Cancer Center, New York, NY 10065, USA; 6Division of Biostatistics, Department of Population and Public Health Sciences, University of Southern California, Los Angeles, CA 90032, USA

**Keywords:** contralateral breast cancer, hereditary breast cancer, exome sequencing, genomics

## Abstract

**Simple Summary:**

As the treatment for breast cancer continues to improve and more women survive their initial diagnosis, there is an increasingly large number of women who are at risk of a second new breast cancer in their lifetimes. However, the hereditary causes of these second breast cancers are not well understood. In this study, we used the latest genetic sequencing technologies to investigate hereditary causes for the second breast cancer in individuals who are known not to have alterations in one of the three main breast cancer genes (*BRCA1*, *BRCA2* and *PALB2*). We analyzed the genetic profiles of selected participants from the WECARE study, one of the largest studies looking at second breast cancers in women. By comparing the genetic profiles of women who have had one breast cancer to similarly matched women who went on to have a second breast cancer, we found that younger women (under 50) with second breast cancers had a higher number of inherited gene alterations compared with those women with one breast cancer. We did not see the same effect in the older women. The results from this study improve our understanding of the hereditary contribution to second breast cancers.

**Abstract:**

Background: Contralateral breast cancer (CBC) is associated with younger age at first diagnosis, family history and pathogenic germline variants (PGVs) in genes such as *BRCA1, BRCA2* and *PALB2.* However, data regarding genetic factors predisposing to CBC among younger women who are *BRCA1/2/PALB2*-negative remain limited. Methods: In this nested case-control study, participants negative for *BRCA1*/*2*/*PALB2* PGVs were selected from the WECARE Study. The burden of PGVs in established breast cancer risk genes was compared in 357 cases with CBC and 366 matched controls with unilateral breast cancer (UBC). The samples were sequenced in two phases. Whole exome sequencing was used in Group 1, 162 CBC and 172 UBC (mean age at diagnosis: 42 years). A targeted panel of genes was used in Group 2, 195 CBC and 194 UBC (mean age at diagnosis: 50 years). Comparisons of PGVs burdens between CBC and UBC were made in these groups, and additional stratified sub-analysis was performed within each group according to the age at diagnosis and the time from first breast cancer (BC). Results: The PGVs burden in Group 1 was significantly higher in CBC than in UBC (*p* = 0.002, OR = 2.5, 95CI: 1.2–5.6), driven mainly by variants in *CHEK2* and *ATM*. The proportions of PGVs carriers in CBC and UBC in this group were 14.8% and 5.8%, respectively. There was no significant difference in PGVs burden between CBC and UBC in Group 2 (*p* = 0.4, OR = 1.4, 95CI: 0.7–2.8), with proportions of carriers being 8.7% and 8.2%, respectively. There was a significant association of PGVs in CBC with younger age. Metanalysis combining both groups confirmed the significant association between the burden of PGVs and the risk of CBC (*p* = 0.006) with the significance driven by the younger cases (Group 1). Conclusion: In younger *BRCA1/BRCA2/PALB2*-negative women, the aggregated burden of PGVs in breast cancer risk genes was associated with the increased risk of CBC and was inversely proportional to the age at onset.

## 1. Introduction

Advances in the treatment of breast cancer (BC) over recent decades have substantially improved the survival rates of women with BC, thereby increasing the population of women at risk of a second primary BC. The risk of developing asynchronous contralateral BC (CBC) is higher than the risk of a first BC in the population [1,2,3], equating to at least 4% at ten years after the first BC [3,4]. The development of CBC is associated with a worse prognosis and other morbidities [5,6]. However, there is no clinical consensus for managing the individual risk of developing CBC, with rates of prophylactic bilateral mastectomies varying significantly across U.S. clinics [7,8,9]. A number of informatic tools estimating individual CBC risk have been deployed to address this challenge [10,11,12,13], but given the lack of data available, their utility and generalizability for accurate clinical decision-making have been limited [13,14].

Risk factors for CBC include young age at the first BC and family history [1,15], indicating that hereditary factors play a role [16]. Treatment for the first BC is also known to modify subsequent CBC risk and has been studied in combination with specific genetic risk factors; examples include the use of tamoxifen in carriers of pathogenic germline variants (PGVs) in *BRCA1* and *BRCA2* [17,18,19] and the effects of radiation exposure in combination with pathogenic variants in *ATM* [20]. Overall, the use of chemotherapy and hormone therapy has been linked to a decrease in CBC risk by 42% [6,21,22].

Pathogenic germline variants (PGVs) in *BRCA1* and *BRCA2* have been associated with an increased risk of CBC [23,24,25]. Carriers of PGVs in *BRCA1* and *BRCA2* are highly susceptible to CBC development with a risk of 2.5% per year after diagnosis of the first BC [24], which is at least five times higher than in non-carriers [2,3]. A number of studies have attempted to identify further PGVs associated with CBC risks, including previous WECARE (for Women’s Environment, Cancer and Radiation Epidemiology) Study publications that analyzed *CHEK2**1100delC and variants in *ATM* and *PALB2* [20,26,27,28]. However, the role of PGVs beyond these genes is not yet known.

The WECARE Study is a population-based case-control study that collected phenotype data and blood DNA samples from women with CBC (cases) and individually matched controls with unilateral BC (UBC) [29]. The rich phenotypic data available from the WECARE Study, including demographic, epidemiologic and clinical information, provide the opportunity to study predisposition to CBC, while adjusting for known general risk factors of CBC. A number of previous publications based on the WECARE Study participants examined the joint roles of environmental risk factors, common variants and rare variants in individual genes, primarily focusing on *BRCA1* and *BRCA2* [18,25,27,30,31,32,33,34]. The present study investigated the aggregated burden of PGVs in established BC genes beyond *BRCA1/2* and *PALB2*, focusing on *BRCA1/2/PALB2*-negative participants in the WECARE Study.

## 2. Materials and Methods

### 2.1. Study Population, Library Preparation and Sequencing of Samples

The participants with CBC (“cases”) and UBC (“controls”) were recruited to WECARE Study as described earlier [29]. Briefly, all cases and controls were diagnosed before age 55 years from 1985 to 2000 with a first primary invasive breast cancer that had not spread beyond regional lymph nodes. Cases were diagnosed with a second primary invasive or in situ CBC at least one year after a first primary diagnosis in 1986 to 2001. Controls were individually matched to each case on birth year, year of first primary diagnosis, cancer registry and race/ethnicity and were required to have an intact contralateral breast. Controls were assigned a reference date reflecting a cancer-free at-risk period following the first BC equivalent to the interval between first and second diagnoses of the matched cases (latency). Cases and controls were required to have had no other prior or intervening cancers, to have resided in a cancer registry catchment area at both diagnoses/reference date, and to have provided a blood sample. Informed consent was obtained from participants. For this study, one control was matched to each case and known carriers of PGVs in *BRCA1/2*/*PALB2* were excluded.

For the current study, participants selected were of European ancestry from the U.S., identified through population-based cancer registries covering Iowa, California (Los Angeles and Orange Counties ) and Washington State associated with the Surveillance, Epidemiology and End Results (SEER) registry system. This study was conducted in two phases using different sequencing techniques (termed “Group 1” and “Group 2”): whole exome sequencing (WES) was conducted in Group 1 while targeted panel sequencing was used for Group 2 using a custom Ampliseq panel that included 106 genes related to BC and DNA repair (Appendix A). Across both groups, participants were over-selected for family history of breast cancer in a first-degree relative, and further matched on the dates of diagnosis and follow-up (latency), race and reporting region. Participants with an early age of onset were prioritized during the 1st group selection. Cases and controls were initially individually matched 1:1; however, as some samples failed sequencing, and some of the individual case-control matching was broken, matching factors were adjusted for the analyses along with cytotoxic chemotherapy, hormonal therapy, breast irradiation, the number of pregnancies and the top PCs, as described below. The models were not adjusted for family history of BC because this could be a proxy for carrying pathogenic germline variants. The age distribution between the two groups was different because the earlier onset cases were selected for Group 1 (Group 1: mean age 42; Group 2: mean age 50).

For both groups, DNA was extracted from blood using QIAGEN columns (QIAGEN LLC, USA, Germantown). For Group 1, the WES libraries (125xPE, 24x multiplexed) were prepared using Illumina Nextera Rapid Exome kits (Illumina, Great Abington, Cambridge, UK; FC−140–9001) following the manufacturer’s recommendations. Sequencing of WES libraries was performed using Illumina HiSeq−2500 machines (Illumina, Great Abington, Cambridge, UK) and SBS v3 or v4 kits (in CRUK Cambridge Institute Genetic core). Each library was sequenced on 4–6 lanes to reach required depth on targets (68 ± 20, mean ± SD). For Group 2, the Ampliseq libraries were prepared and sequenced in SMCL genomics core (Department of Medical Genetics, Cambridge University). The custom library was designed using Thermo Fisher Ampliseq Designer tool (Thermo Fisher Scientific, Waltham, USA). Libraries (150PE, 2 pools) were prepared using a protocol adapted from Konig et al., 2015 [35]. Ampliseq samples were sequenced on 2 lanes of HiSeq 4000 Illumina machines (Illumina, Great Abington, Cambridge, UK) to a mean depth on targets of 442 ± 196 (mean ± SD).

A deidentified study dataset used for the analysis in this publication, which complies with regulatory data sharing restrictions, will be made available upon qualified request to WECARE Study Collaborative Data Repository,

### 2.2. Alignment, Variant Calling and Annotation

The study flowchart is presented in Appendix A. An ethnically close subset of samples (non-Finnish female Europeans, NFFE) from One Thousand Genomes project [36] was used for the joint variant calling with WECARE samples. In addition, these NFFE samples were used for a supplementary supportive analysis to verify that burden of observed PGVs in UBC or CBC was higher than in an unselected population. FASTQ files were demultiplexed and passed through standard QC checks (including FastQC v.0.11.3 and multi-genome alignment [37]). Adaptors and low-quality bases were trimmed using Cutadapt [38]. Alignment and variant calling were performed following GATK Best Practices recommended by the Broad institute at the time of analysis. Reads were aligned to GRCh37 reference genome using BWA MEM (v. 0.7.12) [39]. BAM files from multiple lanes were merged and sorted using samtools (v.1.2) [40]. PCR duplicates were removed in WES, but not in Ampliseq data; diverse QC metrics for alignment and enrichment were then calculated by Picard (https://github.com/broadinstitute/picard accessed on 30 November 2022). FASTQ files for 196 NFFE samples of good quality were selected from One Thousand Genomes project (1KGP) dataset [36]. NFFE FASTQ files were processed by the same pipeline as the WECARE WES data.

Variant calling was performed using GATK (v. 3.4–46 and 3.6–0) [41], and g.vcf files were generated using Haplotype Caller after base quality recalibration and local realignment around indels. Individual g.vcf files were combined into batches of ~100 samples; then, the combined g.vcf-s were used to perform the joint genotyping. Variant calling was performed within the targeted areas only (with 10 bp padding). Variants were filtered by a combination of GATK VQSR and bespoke hard filters, and multiallelic variants were then split to separate lines. GATK down-sampling was suppressed for Ampliseq variant calling and filtering. Variant annotations were added to VCF file using VEP (v.101) [42] and ClinVar (v.20200905) [43]. Along with the standard predicted consequences, the VaP annotations included SIFT (v.5.2.2) [44], PolyPhen (v.2.2.2) [45] and CADD (v.1.6) [46].

Calculations were performed using CRUK CI and Cambridge University high-performance computing clusters. The source data may be requested from the WECARE consortium. The scripts for key steps of the pipeline are available upon request to the WECARE Study Collaborative Data Repository.

### 2.3. Selecting Genes and Pathogenic Variants

The list of BC susceptibility genes was compiled from the studies of Easton et al. (2015), Hu et al. (2021) and Dorling et al. (2021) [47,48,49]. It included *BRCA1/2*, *PALB2*, *ATM*, *CHEK2*, *CDH1*, *TP53*, *PTEN*, *STK11*, *NF1, NBN, RAD51C*, *BARD1* and *RAD51D.* Carriers of PGVs in *BRCA1/2* and *PALB2* were excluded from this analysis. Variants in the remaining 11 genes were analyzed in the WES dataset. *BARD1* and *RAD51D* were not available in the Ampliseq multigene panel, leaving nine BC susceptibility genes for the analysis in the Ampliseq dataset. As the focus of this study was on potentially actionable rare variants with high effect size, rather than on polygenic risks or gene interactions, we used relatively strict criteria in our definition of PGVs:Variants annotated as Pathogenic/Likely pathogenic by ClinVar orLoss of Function variants as predicted by VEP (stop/start gain/loss, frameshift, essential splice sites) orMissense variants that were simultaneously called Deleterious by PolyPhen, Probably Damaging by SIFT and had CADD Phred score > 25.

Of those variants that fulfilled the above criteria, only rare variants (AF < 0.01) with a call rate >0.85 were kept in the analysis. Variants annotated as Benign/Likely benign in ClinVar were excluded.

### 2.4. Statistical Analysis

The burden of PGVs aggregated across the preselected genes taken together was compared in CBC and UBC. Visual assessment of PGVs burden was performed using bar-plots showing aggregated allelic frequencies (AF) and proportions of carriers in the compared groups. The aggregated AF was defined as the total count of ALT alleles (AC) over the mean allele number (AN) across the aggregated variants. The statistical significance of differences between CBC and UBC was evaluated by the Burden test as implemented in the *SKAT* R library (v.2.0.1) [50]. Specifically, *p*-values for dichotomous outcome were calculated using efficient resampling, default (linear weighted) kernel and default settings for imputation and weighting. After excluding variants with call rates <0.85, only a small proportion of genotypes required imputation, where the default SKAT-binary imputation method assigned the most likely values observed in the data (0,1,2) to the missing genotypes. The default weighting implemented in SKAT assigns higher weight to rare variants: numerically the weights are calculated using beta-distribution as dbeta(MAF,1,25). As noted above, covariates added into the model included age at the first CBC, latency, number of pregnancies, use of chemo- and hormonal therapies, and breast irradiation for the first BC. Principal components (PCs) for ancestry were calculated using *bigsnpr* R library (v.1.6.1) [51] based on common variants not in linkage disequilibrium across all genes available in WES/panel data. European ancestry of participants was confirmed by projecting WECARE Study samples to PC space of the major 1KGP populations (Appendix A). Visual assessment of the Scree plots suggested adding the two top PCs as covariates for both Group 1 and Group 2 regression models to account for potential confounding due to residual population stratification (Appendix A). Three top PCs were added to the POLR model in the additional joined analysis of WES + NFFE after evaluation of the Scree plot for the joined WES + NFFE dataset (Appendix A). PCA outliers (if any) were excluded.

Additional subgroup analyses were performed to compare burdens of PGVs in CBC and UBC depending on age or latency (time to CBC) within Group 1 and Group 2 separately. For the age subgroup analysis, study participants were split by median age in each group; the latency threshold for the sub-analysis was set to 5 years. The burden of PGVs within the subgroups was compared using the Fisher test for crude counts and visualized using bar-plots.

Because NFFE controls were unselected for BC, the PGVs prevalence in BC genes is expected to be lower in NFFE than in any WECARE Study sample group (UBC or CBC). Therefore, an additional supportive data check was performed to assess the PGVs burdens between NFFE, UBC and CBC samples. Only variants that had at least 0.85 call rate in both WECARE and NFFE samples were used for this analysis. The NFFE-UBC-CBC trend was visualized by bar-plots for crude counts. Proportional Odds Logit Regression (POLR), as implemented in the *MASS* R package [52], was used for the trend significance assessment (including top PCs into the model).

Metanalysis of summary statistics (*p*-values for PGVs burdens in Group 1 and Group 2) was performed using the METAL package (v.2011–03−25) [53] employing a weighted Z-score to obtain a summary test statistic and *p*-value (i.e., Stouffer’s approach).

## 3. Results

Of the total of 748 samples used for this analysis, 25 samples failed sequencing (15 samples failed WES and 10 samples failed the panel sequencing); 723 samples were successfully sequenced: 334 (172 UBC and 162 CBC) in Group 1 and 389 (194 UBC and 195 CBC) in Group 2.

Table 1 summarizes characteristics of women included in the statistical analysis. The limited sample size precluded analysis of individual variants. At the individual gene level, no genes reached statistical significance after multiple testing correction. In the analysis of PGVs burden over all predefined BC susceptibility genes taken together, the aggregated burden of PGVs in CBC was significantly higher than in UBC in Group 1 (Figure 1A, SKAT Burden test *p* = 0.002). The proportions of PGVs carriers in CBC and UBC in this group were 14.8% and 5.8%, respectively. This was driven mainly by variants in *CHEK2* and *ATM* (Table 2). The full details of PGVs detected in the younger cohort (Group 1, mean age at first breast cancer = 42 years) are given in Appendix A. There was no significant difference in the burden of PGVs between CBC and UBC in Group 2 (mean age at diagnosis of the first BC = 50 years), with proportions of carriers being 8.7% and 8.2%, respectively (Figure 1B, SKAT Burden test *p* = 0.42; details of PGVs detected in Group 2 are given in Appendix A).

### 3.1. Subgroup Analyses by Age and Latency

To further investigate the suggested association with age, we performed a stratified subgroup analysis within each of the studied groups separately, splitting them by the median age at the first BC: 43 years in the younger cohort (Group 1) and 51 years in the older cohort (Group 2). The stratified analysis confirmed higher PGVs burdens in CBC compared with UBC in the younger subgroups of participants (Figure 2).

Additionally, we performed a similar stratified subgroup analysis by latency (time between the first and second BCs). No significant difference was observed between participants who developed the second tumor before and after 5 years from the first tumor (Appendix A).

### 3.2. Metanalysis and Comparison of PGVs Burdens with NFFE

As expected, the burden of observed PGVs in NFFE was lower than in either UBC or CBC (Appendix A), consistent with the fact that 1KGP NFFE represents a population not selected for BC.

Metanalysis combining data from both Groups 1 and 2 confirmed the higher burden of PGVs in CBC compared with UBC (*p* = 0.006, driven by the participants with younger ages at diagnosis). No significant heterogeneity between the groups was detected in the metanalysis (*p* = 0.08).

## 4. Discussion

As BC survival rates improve, there is an increasing need to better understand the risk factors for CBC development. This study measured the aggregated burden of PGVs in established BC genes in *BRCA1/2/PALB2*-negative CBC patients who were part of the WECARE Study. We observed that 14.8% of participants with a younger age at CBC diagnosis (Group 1) carried a PGV in at least one of the studied genes, which was significantly higher than in participants with UBC (5.8%) or in the general population. This excess in the aggregated burden of PGVs was mainly driven by variants in *CHEK2* and *ATM* (Table 2). The absence of similar findings in the older group (Group 2) may be explained by the different biology of BC in women of reproductive age (Group 1: 42 ± 5 years old) and in peri-/postmenopausal age (Group 2: 50 ± 4 years old).

Data on the prevalence of PGVs in CBC beyond *BRCA1/2* and *PALB2* genes are scarce. Yao et al. [54] reported on ~4000 CBC individuals of different ethnicities tested in a single laboratory by panels containing various BC genes. The prevalence of PGVs carriers amongst women with CBC was 3.3% for *CHEK2*, 1.64% for *ATM*, 0.38% for *TP53* and 0.36% for *NBN*. The proportion of *CHEK2* carriers was higher in women with CBC than in women with UBC (*p* < 0.001, driven by individuals of Caucasian origin). However, the study did not have clinical and phenotype data to adjust for treatment, number of pregnancies, and for the fact that some of the first primary BCs, which were used for comparison with CBC, still could develop a second BC later. Fanale et al. (2020) [55] reported results of a panel sequencing for a series of 139 bilateral BCs collected in a single hospital. The study reported 7.9% carriers of PGVs beyond BRCA1/2 and PALB2 genes (five CHEK2, three ATM, two RAD51C and one PTEN variant(s)), which is comparable to the results observed in our study. Of note, the series reported by Fanale et al. [55] included a large proportion of synchronous CBCs (33.1%), and they did not compare the prevalence of PGVs between CBC and UBC or by age at diagnosis.

Our study has several strengths. First, we leveraged a well-characterized study population that allowed the comparison of CBC and UBC, while controlling for multiple known confounding factors. Further, the participants were recruited through population-based cancer registries, rather than in a single hospital series, which enhances generalizability of results. Only early-stage BCs (stages 1 and 2) were included in both UBC and CBC groups, which allowed for longer latency. Including CBC with at least one year between the first and the second cancers ensures that the results predominately apply to the risk of metachronous primary BC. The unique design of the WECARE Study, where UBC controls were individually matched to the CBC cases, as well as the comprehensive phenotype annotations, allowed adjustment for the most important confounding factors influencing the CBC and UBC comparison, such as age at the first BC diagnosis, time to the second BC, treatment, and number of full-term pregnancies.

In spite of these strengths, we were limited by the relatively small sample size: 357 CBC cases and 366 UBC controls altogether, which precluded or limited detailed subgroup analyses or investigation of individual PGVs or genes. This reflects the practical challenges in collecting the dataset where CBC and UBC are matched by age and latency. The limited sample size meant that we had to aggregate variants across multiple genes. Larger BC cohorts will be necessary to assess the individual effects of genes. Two different sequencing techniques were applied to participants with younger and older ages of diagnosis of BC in our study, and while the bioinformatics pipelines were kept as similar as possible, they were not identical.

## 5. Conclusions

In conclusion, we have estimated the burden of PGVs in established BC risk genes in *BRCA1/BRCA2/PALB2*-negative CBC cases compared with UBC controls, and found an increased aggregated burden of PGVs in established BC-risk genes, which was modified by age. These findings provide further evidence of a strong genetic component of breast cancer etiology, beyond the known high penetrance genes among young women, and demonstrate the importance of screening for PGVs in women diagnosed with BC to identify those who are at increased risk of developing CBC.

## Figures and Tables

**Figure 1 cancers-15-00415-f001:**
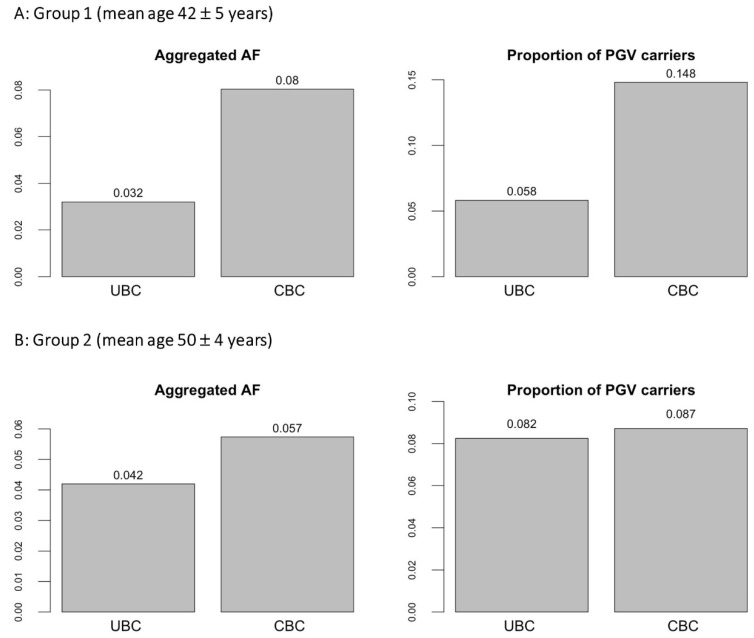
Burden of PGVs in Group 1 and Group 2. Figure notes: In Group 1, the PGVs burden in CBC is significantly higher than in UBC (Burden SKAT test *p* = 0.002). Differences between CBC and UBC in Group 2 are not statistically significant (Burden SKAT test *p* = 0.42). Aggregated ALT allele frequency (aggregated AF) was calculated as the total count of ALT alleles (aggregated AC) over the mean allele number (mean AN) across the aggregated variants. The proportion of PGVs carriers was calculated as the fraction of participants carrying at least one variant in question (homo- or heterozygous). PGV: pathogenic germline variant.

**Figure 2 cancers-15-00415-f002:**
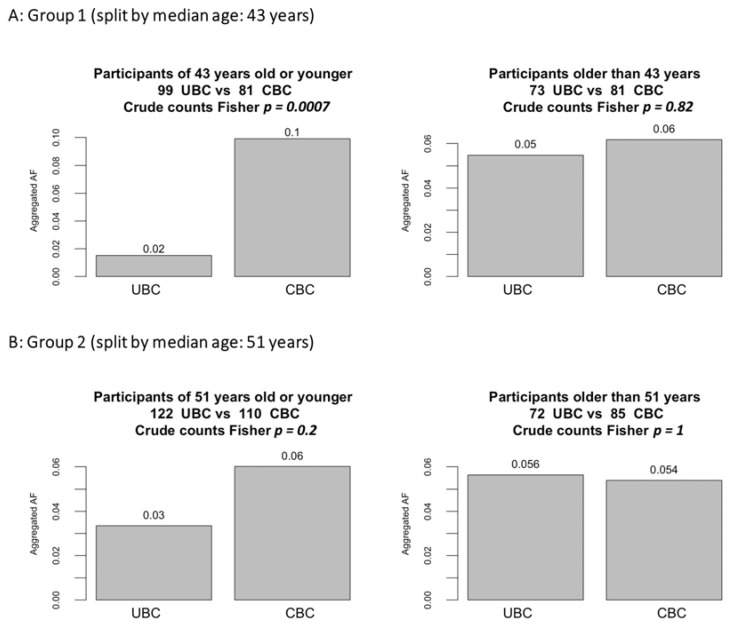
Burden of PGVs stratified by age. Figure note: Each group was split by median age; then, PGVs burdens were evaluated in the subgroups.

**Table 1 cancers-15-00415-t001:** Participant characteristics.

Parameters		Group 1			Group 2		
		UBC	CBC	*p*	UBC	CBC	*p*
Samples Count		172	162	-	194	195	-
Age at the 1st cancer		42 ± 5.5	42 ± 5.4	0.29	50 ± 4.1	50 ± 3.9	0.19
Time to the 2nd cancer or lack of it		6.2 ± 3.5	6.2 ± 3.5	0.88	4.6 ± 2.8	4.6 ± 2.8	0.87
Stage	1	118	119	0.34	143	144	1.00
	2	54	43		51	51	
Estrogen receptors	Neg	47	34	0.31	47	53	0.68
	Pos	92	89		116	108	
	Unknown	33	39		31	34	
Cytotoxic chemotherapy	No	75	83	0.19	99	113	0.19
	Yes	97	79		95	82	
Hormonal therapy	No	115	113	0.55	95	113	0.08
	Yes	56	49		99	82	
	Unknown	1	0		-	-	
Breast irradiation	No	72	89	0.02	63	130	<0.001
	Yes	100	73		131	65	
Number of pregnancies	None	26	38	0.04	32	36	0.78
	1–2	123	113		86	80	
	3 or more	23	11		76	79	
Family history	No	117	97	0.14	133	132	0.98
	Yes	55	65		54	56	
	Unknown	-	-		7	7	

Table notes: *p*-values show significance of differences between contralateral and unilateral breast cancers (CBC and UBC respectively). For continuous data, table shows mean ± std deviation; the significance is estimated by *t*-test. For counts, the significance was estimated by Fisher test for 2 × 2 tables and by Chi square test for 2 × 3 data. CBC: contralateral breast cancer; UBC: unilateral breast cancer.

**Table 2 cancers-15-00415-t002:** PGVs burden per gene in Group 1.

Gene	Variant Count	UBC			CBC		
		Aggregated AC	Mean AN	Aggregated AF	Aggregated AC	Mean AN	Aggregated AF
ATM	11	7	344.0	0.020	10	323.5	0.031
CHEK2	5	2	344.0	0.006	7	324.0	0.022
NF1	2	0	344.0	0.000	2	324.0	0.006
RAD51D	2	2	343.0	0.006	2	324.0	0.006
NBN	1	0	344	0.0000	2	324	0.0062
CDH1	1	0	344	0.0000	1	324	0.0031
TP53	1	0	336	0.0000	1	320	0.0031
RAD51C	1	0	344	0.0000	1	324	0.0031
Total	24	11	343.6	0.032	26	323.6	0.080

Table notes: Aggregated ALT allele count (aggregated AC) was calculated as sum of ALT allele counts over all variants detected in a gene. Mean allele number (mean AN) was calculated as an arithmetic mean of total allele counts for each variant detected in a gene. For instance, if a gene had two variants, and one of them had AN 345 while the other had AN 346, then the mean AN would be 345.5. Aggregated ALT allele frequency (aggregated AF) was calculated as the total count of ALT alleles (aggregated AC) over the mean allele number (mean AN) across the aggregated variants. No pathogenic variants were identified in the other genes studied in Group 1.

## Data Availability

A deidentified study dataset used for the analysis in this publication, which complies with regulatory data sharing restrictions, will be made available upon qualified request to the corresponding author (mdt33@medschl.cam.ac.uk).

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
