# Peer review of "The Contribution of Germline Pathogenic Variants in Breast Cancer Genes to Contralateral Breast Cancer Risk in BRCA1/BRCA2/PALB2-Negative Women"

_cancers, 2023, doi:10.3390/cancers15020415_

Round 1

Reviewer 1 Report

The proposed paper is a useful and original topic that makes the proposed paper attractive, worthy of consideration and it should be of potential interest to a broad readership interested in basic and translational cancer research.

The article is readily understandable because it is well-constructed, clear and well described with figures exhaustive and appropriate to the subject matter. The scientific background, aims are clearly explained and seem to be appropriate to the investigation field.  The topic falls within the scope of the subject area of the journal. In my opinion, this work is acceptable for publication after minor revisions which will help the authors to improve the quality of their manuscript.

-             -   Please replace reference 1 with a more recent reference.

-       -  Please check the acronyms (e.g. line 67 of the introduction). Please check the whole text.

-               -    Please cite and discuss the following article (PMID: 32854451)

          -  What type of breast cancer histology did you consider?

-             -   Please carefully revise the English language throughout the text and correct some grammatical errror and trivial imperfections.

Reviewer 2 Report

The authors aimed to investigate the contribution of germline pathogenic variants (PVGs) at genes that were not BRCA1/BRCA2/PALB2 to Contralateral Breast Cancer. With nested case-control study, the authors found in the young age cohort (mean age at diagnosis 42 years) that PVGs in Contralateral breast cancer (CBC) samples is significantly higher than that in unilateral breast cancer (UBC). But in the old age cohort (mean age at diagnosis 50 years), they did not find such significant differences. They found an increased aggregated burden of PGVs in established BC-risk genes, which was modified by age, demonstrating the importance of screening for PGVs among women diagnosed with BC to identify those who are at high risk of developing CBC. 

Overall, the study designs are perfect and case/control were well applied. For example, they leveraged a well-characterized study population that allowed the comparison of CBC and UBC, while controlling for multiple known confounding factors. The manuscript is well-written, and could have a broad interests in the research field of breast cancer. 

My major concerns are on the methods to select genes and variants in this study.

1. The authors focused on the contribution of PVGs at well-established BC susceptibility genes. How about the contribute of novel genes to CBC susceptibility? The author discussed the limitation of the small sample size in studying the PVG-burden for individual genes. But sounds to me, the authors could conduct PVG-burden analysis for diverse biological pathways, such as KEGG pathway. In each pathway, there are several genes. There should be similar power in pathway gene PVG-burden analysis and your candidate gene PVG-burden analysis. This strategy may overcome the limitation of sample size, and at the same time focused on the contribution novel genes or variants in the development of UBC.

2. Are the criteria for selecting variants too stringent? In the first criteria, Variants annotated as Pathogenic/ Likely pathogenic by ClinVar were selected. As recommended by ACMG/AMP, Pathogenic/ Likely pathogenic variants in ClinVar are interpreted for Mendelian conditions. There are not too many such variants included in ClinVar. Moreover, BC is a complex traits, but not a Mendelian trait. This criteria may exclude a lot of functional important variants in your analysis.
